# The Effects of the Crisis Management Skills and Distance Education Practices of Universities on Student Satisfaction and Organizational Image

Ekrem Toklucu [1,*], Fatoş Silman [1], Selahattin Turan [2], Ramazan Atasoy [3] and Ümit Kalkan [4]

1 Department of Educational Science, Institute of Education and Research, Cyprus International University, Nicosia 99010, Cyprus; fsilman@ciu.edu.tr
2 Department of Educational Administration, Faculty of Education, Uludağ University, Bursa 16059, Turkey; selahattinturan2100@gmail.com
3 Department of Basic Education, Faculty of Education, Harran University, Şanlıurfa 63050, Turkey; atasoyramazan@harran.edu.tr
4 Department of Religious Education, Faculty of Islamic Sciences, Nothern Cyprus Unit, Social Sciences University of Ankara, Nicosia 99010, Cyprus; umit.kalkan@asbu.edu.tr
* Correspondence: etoklucu@gmail.com

**Abstract:** In the present study, the purpose was to determine the direct and indirect effects of the crisis management skills and distance education practices of universities on student satisfaction and organizational image in the continuing Coronavirus pandemic. To conduct the study, a questionnaire was applied to 467 students who had to receive compulsory distance education at TRNC universities during the pandemic process. The relation levels between the crisis management and distance education practices of universities, corporate image, student satisfaction, and direct and indirect effects between the variables, were designed with a structural equation modeling by forming hypotheses according to the sub-dimensions of the student satisfaction scale. The findings of the study showed that as the crisis management of the university administrations in the pandemic process was perceived positively by the students, their organizational image and satisfaction increased. However, it was detected that there was a lower level of relationship between attitudes towards distance learning and crisis management, and that this had a limited effect on student satisfaction. It was concluded that the structural equation model can be used to explain the causal relationship between the variables. The study also showed that the determinants of organizational image and student satisfaction in education must be understood better and that universities must review their crisis management and distance education practices and develop new service plans.

**Keywords:** crisis management; organizational image; student satisfaction; distance education; COVID-19

## 1. Introduction

While today's world is trying to adapt to the changes caused by information and technological developments, the health crisis caused by the new type of Coronavirus pan-demic, whose effects are still ongoing, has undoubtedly been one of the biggest and most important crises in the history of humanity, negatively affecting all sectors, especially education, tourism, and hospitality. With this crisis, which was felt deeply by all people and institutions in the public and private sector, all societies had to face a new normal life order that must be lived with the pandemic, which has become a real danger to all humanity, showing how vulnerable human life is. Humanity, relying on the prevention capacity of information against possible risks [1] and preparing scenarios for crises, has had a hard time coping with this, and was caught unprepared. Although the pandemic caused an additional obstacle to the lives of people struggling with poverty and hunger, it has made trust, social welfare, individual freedom, and social relations more fragile and more manipulated for

people [2,3]. Because COVID-19 is an important, dangerous, and global pandemic with a historical specificity [4,5] unlike any pandemic in the past. COVID-19, which was seen in Wuhan, China at the end of 2019, spread all over the world in a short time and was declared a pandemic by the World Health Organization (WHO) on 11 March 2020 [6,7]. In this respect, traditional (school-based) education was suspended in the Turkish Republic of Northern Cyprus (TRNC) because of the pandemic, and schools were closed at all levels for a certain period in the country to reduce the spread of the pandemic [8–10]. However, this closure lasted longer than expected and continued in the 2020–2021 education period in which decisions were taken not to disrupt the education of higher education institutions in the pandemic, and universities performed their education and training activities with distance education [11]. Thus, there is a lack of study in the literature to show how the pandemic conditions affect the students studying at universities in the TRNC, how the crisis management and distance learning process management of universities were perceived in the process, and how this was reflected on student satisfaction and university image.

This study especially focused on the effects of crisis management skills and distance education practices of universities on student satisfaction and organizational image in the pandemic period. In this respect, to determine the relations between (i) the perceptions of the university students in the TRNC on crisis management and distance education practices, (ii) their satisfaction levels, (iii) the institutional image and student satisfaction, and (iv) to make the universities sustainable. The purpose was to evaluate distance education applications and crisis management skills in the respect of student satisfaction and organizational image. For these purposes, the study was designed with a structural equation modeling the effects of crisis management skills and distance education practices of universities on student satisfaction and the organizational image of universities in the pandemic period.

## 2. Theoretical Foundation

### 2.1. TRNC and Higher Education Sector

The higher education sector in the TRNC tends to grow and develop in agreement with the demand for higher education seen all over the world. Universities are graded into three groups as state, foundation, private sector universities, and additional campus universities opened by state universities in Turkey [12]. According to the latest data from the Ministry of National Education and Culture, 103,108 students received education at universities in the 2020–2021 academic year. The fact that the number of university students, which was around 27,000 at the beginning of the 2000s, has quadrupled in our present-day can be considered a major development for TRNC higher education. It is already known that 45.2% of the students studying in the country are from Turkey, 41.8% are from third world countries and 13% are TRNC nationals, and students come from 136 different countries of the world [13]. In the light of all these data, it can be argued that the number of students who received a university education in the country has great significance both for the sustainability of higher education and for increasing the contribution of the sector to the economic development of the country [11,14]. Working to improve education quality standards in parallel with the number of students, TRNC also tries to produce policies for sustainable higher education. In the scope of sustainable development goals of higher education in sustainable education, it can be predicted that the mobilization of educational resources to contribute to lifelong learning in the types and stages of distance or formal education [15] will have positive effects, especially on student satisfaction. Right at this point, higher education institutions that have advanced crisis management skills can be considered as an important image renewal strategy, with the understanding of providing quality service, their sensitivity to distance education applications, and student satisfaction under all conditions.

Higher education institutions are the service sector providing academic education [16]. In our present day, because of the increasing demand and competition for higher education, meeting the needs and expectations of students is a pressure factor on university

administrations. Although a quality service approach seems to be advantageous in terms of institutional sustainability, there is no certain consensus on service quality since there is no standard for academic quality [17]. Previous research has focused on improving service quality with an internal goal for higher education service providers; however, students' perceptions that dominantly affect university image were ignored while identifying the determinants and results of improving service quality [18–20]. Evidence also revealed that service quality is a multi-dimensional construct. In this regard, Ugboma et al. [21] highlighted these dimensions as tangibles, support services, internationalization, academic staff, and non-academic staff, which affect student satisfaction and organizational image. Considering its integral role in booming competitive advantage and in attracting [22] new and retaining existing students around the world in terms of internationalization, the higher education organizations need to focus on the quality of the organizational image and student satisfaction [23]. Right at this point, how higher education institutions providing academic services are perceived in the eyes of the stakeholders they serve, in other words, their images, can be considered as a parameter that leads to the evaluation of service quality. This perception may emerge especially with the evaluation of student satisfaction. However, despite the positive and significant effect of service quality in particular, academic and non-academic aspects, program issues, university reputation, quality management practices, and access to university facilities on student satisfaction [18,21,24], there are limited studies specifically focusing on the context of mediating the role of distance education practices during the COVID-19 pandemic in the educational administration and leadership (EAL) literature that examines how the services and activities offered by universities affect students' commitment, their perceptions, and satisfaction with the sustainability of the services quality of higher education [18,25–28], the quality of service measured from the perspective of students' commitment, and decision-making processes from the perspective of organizational image [18,21,29].

### 2.2. Crisis Management and Distance Education in Universities

A crisis threatens the short, medium and long-term goals of an institution, sometimes prevents the institution from continuing its life, requires a very urgent response, and creates tension in which the institution's infrastructure for predicting, adapting, and preventing the crisis becomes insufficient [30,31]. The management is the directing of the efforts of individuals of a group in a way that can reach predetermined results in agreement with a common goal by influencing the behaviors and duties of individuals affiliated with a group in a suitable environment [32]. Crisis management must be sustainable. In effective crisis management, it is essential to evaluate the process after the crisis ends and returns to normal order. This evaluation can be considered a guiding guide for organizations to develop their crisis management skills. Especially in this period that has not been overcome yet and is called the new normal, the increased severity of the crisis still maintains its current status. Quick and accurate decisions taken in the early stages of the crisis are extremely important for both corporate sustainability and corporate image. In such crisis periods, this situation has special importance for higher education institutions [33]. Because higher education institutions face effects that increase, develop, and spread sustainability concerning the formation of human capital, the construction of knowledge bases, the dissemination, use, and maintenance of knowledge is important. In the construction of a sustainable future, universities have a mission that distinguishes them from other organizational structures, which is essentially the phenomenon of informing the people and raising awareness in society [34,35].

For this reason, especially in times of crisis, the eyes of the public focus on universities that produce academic knowledge, and the crisis management skills put into practice are frequently discussed by the stakeholders. In this respect, the university administration must be prepared in advance for the crisis and its consequences, take the necessary precautions, activate the early warning systems, successfully control the chaos caused by the crisis, manage the crisis with the least damage and return to a normal state [30,36]. The most

important secret for being able to overcome the crisis with the least damage and manage the chaos caused by the crisis in a healthy way is to feel the signals of the crisis in advance and to calculate the possible consequences correctly. In other words, the preparation of crisis action plans and programs before the crisis occurs is an important management approach. The studies made on the crisis management competencies of universities during the pandemic show that it is determined by the attitudes and behaviors of higher education institution administrators and lecturers about digitalization rather than the lack of digital infrastructure [37].

It is already known that it is important for universities to conduct their educational activities effectively by moving from the theoretical structure of distance education and to ensure the sustainability of higher education studies with and without interruption in times of crisis. For this reason, in the process we live in, it is a great necessity to make full use of information and information technology and to support education and training environments with new methods and techniques. Right at this point, the significance of establishing an effective crisis management system by using crisis management skills of universities emerges once again. Additionally, it has great significance to evaluate the crisis management skills and distance education applications used by universities in the current pandemic crisis period and to determine the trends in institutions. In this way, it has great significance for universities to take a strategic position in agreement with the wishes, suggestions, and expectations of students and other stakeholders by following the developments on a global scale, and to run their educational studies with a universal and sustainable perspective. This emphasizes that distance education activities performed to gain intense competitive power within national and transnational borders must be evaluated in terms of student satisfaction and corporate image. In this respect, universities must be reminded that they must identify the problems they encounter in planning distance education activities and act with different management strategies and scenarios in future crisis periods to cope with these problems. All taboos related to the traditional understanding of education were broken in the pandemic process, and habits were replaced by the reshaping of the education system and the birth of new paradigms [38]. It is predicted that distance education, which has been tested in academic fields, may be a necessity for information and technology societies of the final century and an understanding of education in crisis processes [39]. It is predicted that the traditional education approach will be insufficient to meet the needs of future generations. With regards to all these discussions, the effectiveness, efficiency, quality, advantages, and disadvantages of distance education must be examined. Without a doubt, making these examinations in the perspective of student opinions, which is the important output of education, will be considered an important step in achieving the most reliable results [40].

### 2.3. Student Satisfaction and Organizational Image

Higher education institutions, which understand how important student satisfaction and perception are in our present day's information society, emphasize the significance they attach to student satisfaction in their mission documents, assumptions, and promotional activities [41]. Student satisfaction is expressed as, "the subjective and positive evaluations of students about various aspects and outcomes related to the education they receive in their institution" [42]. The fact that student satisfaction, which is a multidimensional and complex concept, is an educational output on its own, has turned this concept into one of the topics of interest in a study for a long time [41,43]. It is considered one of the important principles of quality standards in higher education. In accordance with customer orientation, the educational service received by students, who are the most important customers of higher education, and the satisfaction they provide from it is extremely valuable for increasing the quality of the institution [44,45]. All higher education institutions of the state, foundation, and private sector must satisfy their students with this educational service they offer and make this satisfaction sustainable [21]. Similar opinions are expressed in studies on the subject. Elliott and Shin [41] underlined the need to focus

on student satisfaction with the quality of service, to develop a mechanism that allows the administration to design their institutions following the requirements of the age to adapt to the needs and expectations of the students and to develop a mechanism that allows needs analysis and needs satisfaction assessment to be performed at regular intervals. Additionally, it is argued that student satisfaction is an important aspect of achieving a sustainable competitive advantage in the higher education industry. In this direction, Khosravi et al. [46] emphasized in their study that meeting the needs and expectations of students is an imperative duty for higher education institutions to gain a sustainable competitive advantage against their competitors.

The desire to enroll in a higher education institution and the priority of choosing that specific institution emphasize the quality and image of that institution [47]. Image has started to occupy an important place in the educational market, and in the management of the services offered, this concept is heavily used as a parameter effective in the preference of students for a higher education institution [48]. Whether a service is provided to meet the expectations of the students indicates the satisfaction of the students with their universities [47]. Student satisfaction can also be examined with a different understanding of the field. In this respect, it was reported that there is competition in the educational sector as in all other sectors, marketing strategies are involved, and student satisfaction, like the image, is a part of these strategies [49]. Additionally, studies show that service quality and image are among the variables that affect student satisfaction, and it is suggested that these variables predict each other. For this reason, it is recommended that universities conduct activities that improve their academic staff, internationalization, and image to increase student satisfaction [21].

The image concept, which was used for the first time by Levy, can be defined as all of the impressions, judgments, and thoughts about a person, institution, or organization people see and perceive [50]. Organizational image is, "the beliefs and feelings of customers/stakeholders about an institution" [51]. The fact that an institution knows what kind of image perception exists in its environment can be considered feedback on its services and activities. With these, the institution understands how it is perceived and does not act blindly [52]. Opportunities and services offered by the organization also play a major role in the perception of corporate image [53]. At the turning point of the 21st century, the concept of "image" is considered a central concept as a common field of study in governance science. In the focus of modern-style social criticism, it is emphasized that we live in a society saturated with images. On the other hand, Christensen and Askegaard [54] state that image creation is one of the most important issues for organizations and marketing experts, and institutions as the building blocks of the social and financial sector.

Organizational structures were exposed to an open structure and intense competition with globalization. In education, which is considered a service namely marketed worldwide, the higher education sector has also become globalized. Higher education institutions provide student inputs from within the country and from outside the borders of the country. Right at this point, universities must compete with each other to attract high-quality students and academic staff at an international level. Organizational image is an important resource in such a global market environment and a positive image will attract customers to the organization [55,56]. Thus, the academic, cultural and social development levels of universities are an important perception factor. However, globalization was not able to overcome the developmental problem of educational organizations relative to each other. Higher education institutions that cannot keep up with the developments required by the age and even leaders in these innovations cannot achieve the desired success. At this stage, as in Welford's [57] institutional sustainability model, universities must redefine their organizational identities by updating their policies and goals if necessary, and continue their changes and achieve their goals after making these images consistent with the sustainable development phenomenon.

The findings obtained as a result of the present study have great significance in terms of evaluating the distance education practices and institutional images of TRNC

universities and showing student satisfaction in the crisis caused by the global pandemic. In fact, distance education applications have been the subject of research in the literature for years. Researchers have focused on trying to determine positive or negative parameters of distance learning on the new models related to how individuals can educate themselves, the pedagogy of distance education, its impact on disadvantaged groups, and the determination of institutional and government policies by indicating future research needs, and the global, political, economic, and technological pressures on distance education [58–64]. Evidence demonstrated that there is a significant relationship between distance education practices and student satisfaction. However, there are uncertainties about the future of distance education and its sustainability all over the world, especially related to the problems in the access to resources for disadvantaged students. In addition to this, the researchers pointed out that the views of stakeholders were not included in the restructuring process regarding distance learning, there were some negative effects of distance learning in terms of pedagogy, and the level of student satisfaction was low due to the problems related to the insufficient level of technology integration by the instructors [65–70]. The reason for this is that education is one of the most affected social areas in the pandemic process and it is important to be aware of the possible negative effects as soon as possible. It was observed, especially with the distance education applications faced with the pandemic crisis, that many parameters respecting student satisfaction have changed [69], and satisfaction indicators were viewed from a different perspective. Student satisfaction is a dynamic concept [71]. This can change in coordination with the conditions of the world, innovations, and developments. Right at this point, it reveals the necessity of universities to integrate with the changes and innovations that take place at a dizzying speed, they must manage crises, and they must measure student satisfaction in a sustainable way [72]. Showing the opinions and satisfaction of students on the distance education practices and crisis management skills of universities has a critical significance in determining the advantages and disadvantages of distance education, which is faced for the first time in such a crisis period. The study is also important for TRNC higher education institutions to realize student satisfaction and establish a more student-oriented university education identity. In the literature, no research was found focusing on the organizational image and student satisfaction of distance learning within the scope of the emergency crisis action plan, especially in higher education institutions, in the most referenced indexes in the field of EAL (Web of Science, Scopus, and Eric). Right at this point, the findings obtained in terms of the added value it will provide to the literature are also very important. Although the direct relationships between student satisfaction and organizational image seem comprehensible, the evidence for universities coping with difficulties through distance education in times of crisis remains unclear. The results obtained here will fill this gap in the literature by shedding light on the policies to be developed for universities and the regulations of university administrations.

*2.4. Conceptual Respect*

During the new "normal", which emerged in the COVID-19 pandemic, this study was conducted to uncover the direct and indirect effects of crisis management [CM] skills of TRNC higher education institutions on distance learning [DL] attitudes, students satisfaction sub-dimensions (social and cultural satisfaction [SCS], satisfaction with R&D activities [RDS], satisfaction with quality monitoring [QMS], satisfaction with the instructional process [INS], satisfaction with the design of instruction [IDS], satisfaction with the instructional environment and resources [IES]) and organizational image [OI] perceptions. The conceptual respect of the model created for this purpose is given in Figure 1.

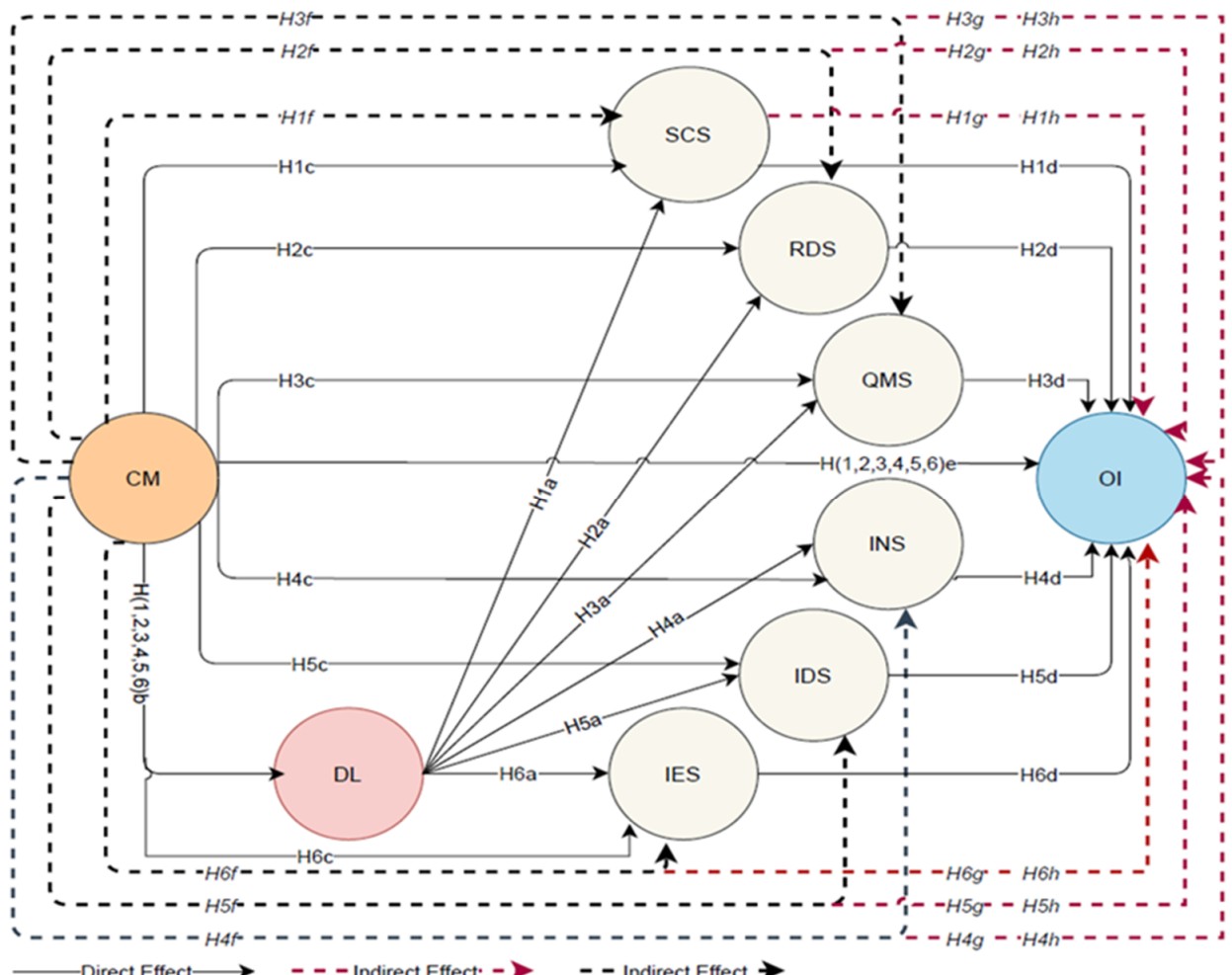

**Figure 1.** Conceptual Model.

The conceptual respect of this study is derived from several decades of theoretical and empirical studies in the field of educational administration. Oliver [73], Thomassen [74], and Tang and Wang [75] discussed satisfaction concerning, "the customer's perception, consciously or unconsciously, as a result of comparing their face with their expectations". Kotler and Keller [76] defined customer satisfaction as, "the degree to which a person is satisfied or disappointed with the observed performance of a product in line with expectations". Related satisfaction studies focusing on customer (student) satisfaction theory [74,75,77] in the literature form the basis for the theoretical respect of the present study. In the literature, satisfaction was detected to be associated with many internal and external factors, and it was reported in previous studies that the performance of the organizations (universities) below the expectations is because of the low level of satisfaction of the customers (students), a performance that meets or exceeds expectations leads to a high level of satisfaction in the relevant literature. In the study, organizational image, which is among the reflected parameter indicators of customer satisfaction [78], was constructed as a second dependent variable. In this respect, the study questions/hypotheses (Figure 1) created respecting the crisis management put forward by the university administrations in the pandemic process of the students who had to receive distance education compulsorily in the study, affect their perceptions of satisfaction with their universities and their organizational image perceptions directly or indirectly (Figure 1).

**Hypotheses:**

1. Hypotheses regarding the direct and indirect effects of CM on DL and the six dimensions of student satisfaction: *What are the direct and indirect effects of the crisis management of universities on the distance education and six sub-dimensions of student satisfaction*[$H^{CM, DL, SS}$ (1, 2, 3, 4, 5, 6) a, b, c, f ]?

2. Hypotheses regarding the direct and indirect effects of CM on DL and the six dimensions of student satisfaction: *What are the direct and indirect effects of the crisis management of universities on the six sub-dimensions of student satisfaction and organizational image* [$H^{CM, SS, OI}$ (1, 2, 3, 4, 5, 6) d, e, g]?

3. Hypotheses regarding the indirect effects of CM on OI and via the distance education and six sub-dimensions of student satisfaction: *What are the direct and indirect effects of the crisis management of universities on the distance education and six sub-dimensions of student satisfaction* [$H^{CM, DL, SS, OI}$ (1, 2, 3, 4, 5, 6) h]?

## 3. Data and Methodology

The study was characterized by the collection of data at a specific time in the COVID-19 pandemic. For this reason, a cross-sectional explanatory design was used. In cross-sectional studies, there is no time dimension related to the study subject, and the data are collected and usually refer to the time of data collection or the time frame around it [79]. In this case, the year 2021, when the pandemic process continued, was evaluated as the period of May. In this process, a descriptive study was performed in the relational questionnaire model to examine the direct and indirect relations between the crisis management skills of universities, student satisfaction with distance education applications, and organizational image [80].

### 3.1. Sampling

The sampling of this study consisted of students studying at 10 public and private universities in the TRNC. The reason for determining the TRNC for the research is that the number of students in higher education is high and it is one of the sectors that contribute to the country's economy together with tourism [TRNC Population (2019): 382.230/Student Population (2021): 103.108] [81]. A total of 467 questionnaires were considered valid for data analysis since 519 students participated in the study and 52 students' questionnaire data were missing or not correct. The data were collected from the students at Cyprus International University, European University of Lefke, Girne American University, METU Northern Cyprus Campus, Near East University, ASBU Northern Cyprus Academic Unit, Final International University, Eastern Mediterranean University, University of City Island, and ITU North Cyprus using a questionnaire conducted online in May 2021. While determining the universities to be included in the research universe, students from 10 universities were selected that met the inclusion criteria. The research tool has been converted into an electronic format, which the participants could fill out online. The online form was sent to all students. Participation in the study was completely voluntary. In the prepared form, demographic information such as university, faculty, and department were taken apart from the items, and no information was requested that would reveal the identity of the participant. In order to make the created form available to more users, it was kept open to the access of the participants for a period of one month and access to the form was closed at the end of the period.

To support a better understanding of the study data, the sampling size, a detailed description of the universities, and the frequency and percentages of demographic factors, e.g., gender, age, grade, and departments of the participants are given in Table 1.

### 3.2. Study Tools

The measurement tools were created by combining four different measurement tools with personal information to determine the crisis management skills of universities and

the perceptions of student satisfaction, distance education applications, and organizational image. Details of each dimension are summarized below.

**Table 1.** Sampling Characteristics.

| Variables | | N | % |
|---|---|---|---|
| Universities | Cyprus International University | 94 | 20.1 |
| | European University of Lefke | 72 | 15.4 |
| | Girne American University | 67 | 14.4 |
| | METU Northern Cyprus Campus | 45 | 9.6 |
| | Near East University | 41 | 8.8 |
| | ASBU Northern Cyprus Academic Unit | 37 | 7.9 |
| | Final International University | 32 | 6.9 |
| | Eastern Mediterranean University | 30 | 6.4 |
| | University of City Island | 25 | 5.3 |
| | ITU North Cyprus | 24 | 5.2 |
| | Total | 467 | 100 |
| Citizen | North Cyprus | 121 | 26 |
| | Turkey | 322 | 68.9 |
| | Others | 24 | 5.1 |
| | Total | 467 | 100 |
| Gender | Male | 193 | 41.3 |
| | Female | 274 | 58.7 |
| | Total | 467 | 100 |
| Age | 16–18 | 112 | 24 |
| | 19–21 | 280 | 60 |
| | 22 and above | 75 | 16 |
| | Total | 467 | 100 |
| Class | Preparatory class | 13 | 2.8 |
| | 1. Grade | 50 | 10.7 |
| | 2. Grade | 117 | 25.1 |
| | 3. Grade | 182 | 39 |
| | 4. Grade and above | 105 | 22.5 |
| | Total | 467 | 100 |
| Faculty | Education | 143 | 30.6 |
| | Engineering/Architecture | 53 | 11.3 |
| | Law | 52 | 11.1 |
| | Nursing | 46 | 9.9 |
| | Science–Literature | 35 | 7.5 |
| | Tourism–Hotel | 35 | 7.5 |
| | Pharmacy | 34 | 7.3 |
| | Medicine | 31 | 6.6 |
| | Dentistry | 14 | 3.0 |
| | Business Management/Economy | 14 | 3 |
| | Computer and Technology | 10 | 2.1 |
| | Total | 467 | 100 |

1. Crisis Management [CM] Dimension: This dimension focused on the crisis management skills of universities when the pandemic process first emerged. The level of perception of the crisis signals of universities before the pandemic included the strategies to create crisis scenarios for education and training, detect problems that may pose danger, examine every aspect that may cause a crisis, be sensitive to the signs of crisis, and protect against the negative effects of the crisis. Seven items of the precrisis period dimension ($\alpha = 0.95$) of the scale developed by Aksu and Deveci [82] were used in the study. The Cronbach Alpha value of the overall scale was ($\alpha = 0.98$).

2. Distance Education [DL] Dimension: This dimension focused on the distance education opportunities offered by universities to students during the pandemic process. The scale was developed by Arslan [83] and consisted of 21 items and 5 sub-dimensions to investigate the competencies of faculty members in distance education, student attitudes towards online exams, comparison of distance education and face-to-face education, communication, and access to distance education. Satisfaction with the opportunities offered by the university in distance education ($\alpha = 0.87$), attitude towards faculty members in distance education ($\alpha = 0.89$), attitude towards online exams ($\alpha = 0.79$), communication and access in distance education ($\alpha = 0.65$), distance education, and comparison of face-to-face education ($\alpha = 0.68$), and the Cronbach Alpha value of the overall scale was 0.88. The analyzes were made on the general dimension of the distance education measurement tool in the study. It was detected that the DFA values used to test the construct validity of the distance education scale data set indicate a Goodness of Fit Index in terms of model fit ($\chi^2 = 360.686$, df = 179, $\chi^2/df = 2.015$, RMSEA = 0.047, RMR = 0.027, GFI = 0.930, AGFI = 0.910, IFI = 0.974, TLI = 0.969, CFI = 0.974).

3. Organizational Image [OI] Dimension: This dimension focuses on the organizational image of universities in the pandemic period. In this process, its purpose was to determine the quality perception of the students to prepare the students of the universities for academic life and the business world and to meet all kinds of needs of the students with 5 items. These items are described in Kazoleas et al. [84] and Polat et al. [85]. The quality image of the scale was Cronbach Alpha value ($\alpha = 0.90$), and the Cronbach Alpha value for the overall scale was ($\alpha = 0.92$).

4. Student Satisfaction [SS] Scale: This scale is the final dimension consisting of 45 items to obtain information about the general satisfaction level of students at their university. The scale was developed by Şimşek, Islim, and Öztürk [86] and consists of 6 dimensions. Satisfaction with the design of instruction [IDS] ($\alpha = 0.84$), satisfaction with the instructional process [INS] ($\alpha = 0.88$), satisfaction with quality monitoring [QMS] ($\alpha = 0.91$), study and development activities in the scale satisfaction with R&D activities [RDS] ($\alpha = 0.91$), satisfaction with social and cultural activities [SCS] ($\alpha = 0.93$), and satisfaction with the instructional environment and resources [IES] ($\alpha = 0.91$), Cronbach's Alpha value for the overall scale was greater than 0.70. It was seen that the DFA values performed to test the construct validity of the student satisfaction scale dataset indicated a Goodness of Fit in terms of model fit ($\chi^2 = 1559.929$, df = 930, $\chi^2/df = 1.677$, RMSEA = 0.038, RMR = 0.020, GFI = 0.872, AGFI = 0.858, IFI = 0.977, TLI = 0.975, CFI = 0.977). CFA results show that the Goodness of Fit Indexes respecting the structure of the scales is at an acceptable level in terms of study data [87].

*3.3. Analysis of Data*

The study data were analyzed by using the SPSS 25 package program, IBM Amos 22 software, and Mplus 8.3. However, before starting the analysis of the dataset, it was determined whether there were contrary values in the data by using frequency values and Mahalanobis distances. In the analysis, firstly, the validation status of the measurement model respecting the variables given in the model was tested. In the next step, the predictions related to the model were analyzed using the Mplus 8.3 program [88].

## 4. Findings and Interpretation

*4.1. Analysis Findings for Crisis Management, Distance Education, Student Satisfaction Sub-Dimensions, and Organizational Image*

The Pearson Product-Moment Correlation Analysis was conducted to determine the relations between the variables and explanatory statistics to evaluate the crisis management, organizational image, distance education, and satisfaction levels of the students. In this respect, the results are given in Table 2.

According to Table 2, it is seen that the reliability of the scales that measure each variable is well above the limit [$\alpha = 0.70$] which is accepted as the limit in many studies.

However, according to descriptive statistics, it was detected that the lowest average is distance learning, and the highest average is crisis management and organizational image perception, respectively. In addition, when the relation coefficients between the variables were examined, a positive relation was detected between all the variables in a spectrum [0.290/0.712] varying in the low, medium, and high levels. In this respect, the crisis management of the university administrations during the COVID 19 pandemic process means that the organizational image and satisfaction levels increase as the level of perception is perceived positively by the students.

**Table 2.** Descriptive Statistics and Pearson Relations between Variables.

|  | **Mean** | **SD** | **α** | **OI** | **SCS** | **RDS** | **QMS** | **INS** | **IDS** | **IES** | **DL** |
|---|---|---|---|---|---|---|---|---|---|---|---|
| CM | 3.64 | 0.70 | 0.92 | 0.615 * | 0.700 * | 0.498 * | 0.497 * | 0.448 * | 0.429 * | 0.555 * | 0.488 * |
| OI | 3.50 | 0.57 | 0.86 | 1 | 0.570 * | 0.377 * | 0.382 * | 0.343 * | 0.370 * | 0.463 * | 0.381 * |
| SCS | 3.18 | 0.64 | 0.96 |  | 1 | 0.614 * | 0.615 * | 0.573 * | 0.529 * | 0.590 * | 0.363 * |
| RDS | 3.11 | 0.76 | 0.94 |  |  | 1 | 0.660 * | 0.674 * | 0.594 * | 0.671 * | 0.309 * |
| QMS | 3.21 | 0.78 | 0.93 |  |  |  | 1 | 0.694 * | 0.578 * | 0.631 * | 0.344 * |
| INS | 3.27 | 0.75 | 0.96 |  |  |  |  | 1 | 0.713 * | 0.712 * | 0.299 * |
| IDS | 3.09 | 0.86 | 0.98 |  |  |  |  |  | 1 | 0.697 * | 0.290 * |
| IES | 3.32 | 0.77 | 0.93 |  |  |  |  |  |  | 1 | 0.335 * |
| DL | 2.96 | 0.59 | 0.94 |  |  |  |  |  |  |  | 1 |

Notes: n = 467; SD: Standard Deviation, α: Cronbach Alpha; * *p* < 0.01.

The relation values between all variables were detected below 0.80, and then structural equation model analysis was performed to determine the relations between dependent variables and independent variables. In this respect, the parameter estimates for the model are given in Figure 2.

*4.2. Model 1 for Crisis Management, Distance Education, Social and Cultural Satisfaction, and Organizational Image*

The model that was created for the direct and indirect effects of universities' crisis management skills, distance education, social and cultural satisfaction, and organizational image perceptions according to student perceptions is given in Table 3.

It is seen in Table 3 that all Goodness of Fit Indices of the model established was quite good. When Table 3 and Figure 2 are examined, it is seen that DL has SCS [β = 0.43; S.E: 0.04; t: 10.51; *p* < 0.001; H1a], DL of CM [β = 0.31; S.E: 0.05; t: 6.72; *p* < 0.001; H1b] and CM, SCS [β = 0.35; S.E: 0.04; t: 8.82; *p* < 0.001. It appears to have a moderately positive direct effect on H1c]. In this respect, a one-unit increase in DL has a 0.43-unit increase on SCS; a one-unit increase in CM indicates an increase of 0.31 in DL and 0.35 in SCS. Similarly, OI of SCS [β = 0.36; S.E: 0.04; t: 9.81; *p* < 0.001; H1d] and CM also have OI [β = 0.56; S.E: 0.03; t: 16.82; *p* < 0.001; H1e] as positive and moderate direct effects. In other words, a one-unit increase in SCS indicates a 0.36 increase in OI, and a one-unit increase in CM indicates a 0.56 increase in OI.

In the established model, an indirect effect of CM on SCS via DL was detected [β = 0.13; S.E: 0.023; *p* < 0.001; H1f]. Similarly, an indirect effect of CM on OI via SCS was detected [β = 0.125; S.E: 0.02; *p* < 0.001; H1g]. Another indirect specific effect was detected on the OI of CM via the SCS and DL pathway [β = 0.047; S.E: 0.010; *p* < 0.001; H1h].

*4.3. Model 2 Created for Crisis Management, Distance Education, Satisfaction with R&D Activities, and Organizational Image*

Model 2, which was created for the direct and indirect effects of universities' crisis management skills, satisfaction with distance education, satisfaction with the R&D activities, and organizational image perceptions according to student perceptions, is given in Table 4.

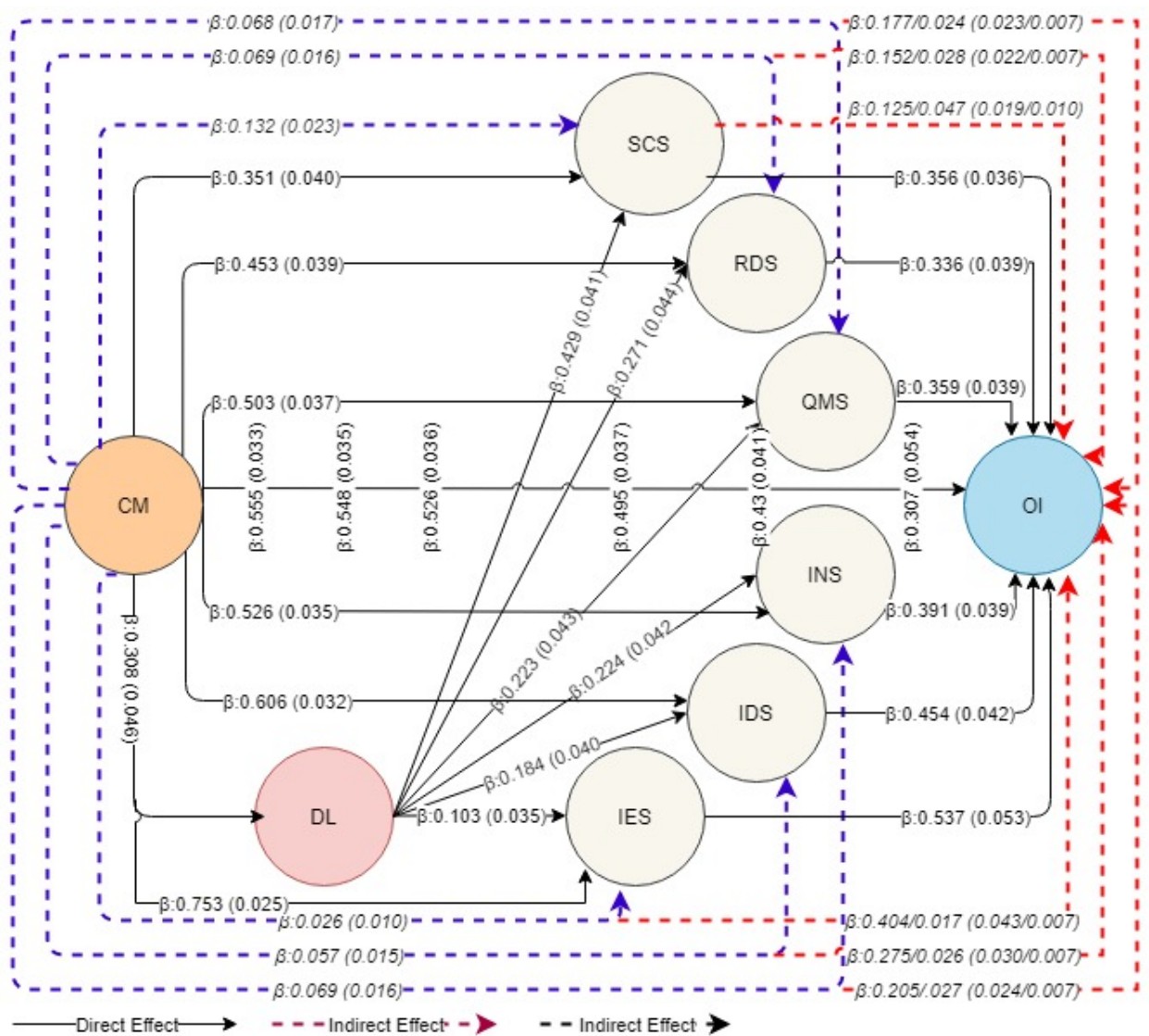

**Figure 2.** Standardized Direct, Indirect, and Total Effects of All Hypotheses.

**Table 3.** Standardized Direct, Indirect and Total Effects on Model 1.

| Hypotheses | Direct Effects | | | Indirect Effect 1 | | | Indirect Effect 2 | | | TE |
|---|---|---|---|---|---|---|---|---|---|---|
| SCS→H1a/b/c/d/e/f/g/h | β | SE | *p* * | β | SE | *p* * | β | SE | *p* * | β |
| H1a:[DL→SCS] | 0.429 | 0.041 | 0.001 * | | | | | | | |
| H1b:[CM→ DL] | 0.307 | 0.046 | 0.001 * | | | | | | | |
| H1c:[CM→SCS] | 0.351 | 0.040 | 0.001 * | | | | | | | |
| H1d:[SCS→OI] | 0.356 | 0.036 | 0.001 * | | | | | | | |
| H1e:[CM→OI] | 0.555 | 0.033 | 0.001 * | | | | | | | |
| H1f:[CM→DL→SCS] | | | | 0.132 | 0.023 | 0.001 * | | | | 0.482 |
| H1g:[CM→SCS→OI] | | | | 0.125 | 0.019 | 0.001 * | 0.047 | 0.010 | 0.001 * | 0.727 |
| H1h:[CM→DL→SCS→OI | | | | | | | | | | |
| MFI | CFI | TLI | RMSE | SRMR | | | $\chi^2$/Sd | | | |
| MFI Values | 0.949 | 0.945 | 0.048 | 0.045 | | | 1833.13/892 = 2.055 | | | |

* *p* < 0.001; MFI: Modification indices; TE: Total effect.

**Table 4.** Standardized Direct, Indirect and Total Effects of Model 2.

| Hypotheses | Direct Effects | | | Indirect Effect 1 | | | Indirect Effect 2 | | | TE |
|---|---|---|---|---|---|---|---|---|---|---|
| RDS→H2a/b/c/d/e/f/g/h | β | SE | *p* * | β | SE | *p* * | β | SE | *p* * | β |
| H2a:[DL→ RDS] | 0.271 | 0.044 | 0.001 * | | | | | | | |
| H2b:[CM→ DL] | 0.308 | 0.046 | 0.001 * | | | | | | | |
| H2c:[CM→ RDS] | 0.453 | 0.039 | 0.001 * | | | | | | | |
| H2d:[RDS→ OI] | 0.336 | 0.039 | 0.001 * | | | | | | | |
| H2e:[CM→ OI] | 0.548 | 0.035 | 0.001 * | | | | | | | |
| H2f:[CM→ DL→ RDS] | | | | 0.083 | 0.018 | 0.001 * | | | | 0.536 |
| H2g:[CM→ RDS→ OI] | | | | 0.152 | 0.022 | 0.001 * | 0.028 | 0.007 | 0.001 * | 0.728 |
| H2h:[CM→ DL→ RDS→ OI] | | | | | | | | | | |
| MFI | CFI | TLI | RMSE | SRMR | | | $\chi^2$/Sd | | | |
| MFI Values | 0.976 | 0.974 | 0.039 | 0.045 | | | 1018.95/619 = 1.646 | | | |

* *p* < 0.001; MFI: Modification Indices; TE: Total Effect.

According to Table 4, it can be argued that all Goodness Of Fit Indices in the established model have quite good values. When Table 4 and Figure 2 are examined, it is seen that DL has RDS [β = 0.27; S.E: 0.04; t: 6.18; *p* < 0.001; H2a], DL of CM [β = 0.31; S.E: 0.05; t: 6.74; *p* < 0.001; H2b] and CM, RDS [β = 0.45; S.E: 0.04; t: 11.74; *p* < 0.001; a positive direct effect on H2c] was detected. In this respect, a one-unit increase in DL has a 0.27-unit increase on RDS; a one-unit increase in CM indicates an increase of 0.31 in DL and 0.45 in RDS. Similarly, OI of RDS [β = 0.37; S.E: 0.04; t: 8.72; *p* < 0.001; H2d] and CM also have OI [β = 0.55; S.E: 0.04; t: 15.61; *p* < 0.001;H2e] as positive moderate direct effects. This finding indicates a 0.37 increase in OI for a one-unit increase in RDS and a 0.55 increase in OI for a one-unit increase in CM.

An indirect effect of CM on RDS via DL was determined in the model [β = 0.08; S.E: 0.02; t: 4.54; *p* < 0.001; H2f]. Similarly, an indirect effect of CM on OI via RDS was also detected [β = 0.15; S.E: 0.02; t: 7.06; *p* < 0.001; H2g]. Another indirect specific effect was detected on the OI of CM via the RDS and DL pathway [β = 0.03; S.E: 0.010; t: 3.97; *p* < 0.001; H2h].

*4.4. Model 3 Created for Crisis Management, Distance Education, Monitoring, Evaluation, and Quality Management Satisfaction and Organizational Image*

Model 3, which was created for the direct and indirect effects of universities' crisis management skills, distance education and monitoring, evaluation and quality management, and organizational image perceptions according to student perceptions, is given in Table 5.

**Table 5.** Standardized Direct, Indirect and Total Effects on Model 3.

| Hypotheses | Direct Effects | | | Indirect Effect 1 | | | Indirect Effect 2 | | | TE |
|---|---|---|---|---|---|---|---|---|---|---|
| QMS→H3a/b/c/d/e/f/g/h | B | SE | *p* * | β | SE | *p* * | β | SE | *p* * | β |
| H3a:[DL→QMS] | 0.221 | 0.043 | 0.001 * | | | | | | | |
| H3b:[CM→ DL] | 0.308 | 0.046 | 0.001 * | | | | | | | |
| H3c:[CM→QMS] | 0.503 | 0.037 | 0.001 * | | | | | | | |
| H3d:[QMS→OI] | 0.352 | 0.039 | 0.001 * | | | | | | | |
| H3e:[CM→OI] | 0.526 | 0.036 | 0.001 * | | | | | | | |
| H3f:[CM→DL→QMS] | | | | 0.068 | 0.017 | 0.001 * | | | | 0.571 |
| H3g:[CM→QMS→OI] | | | | 0.177 | 0.023 | 0.001 * | 0.024 | 0.007 | 0.001 * | 0.728 |
| H3h:[CM→DL→OI→QMS] | | | | | | | | | | |
| MFI | CFI | TLI | RMSE | SRMR | | | $\chi^2$/Sd | | | |
| MFI Values | 0.937 | 0.933 | 0.057 | 0.056 | | | 1952.88/769 = 2.54 | | | |

* *p* < 0.001; MFI: Modification Indices; TE: Total Effect.

When Table 5 is examined, it is seen that all Goodness Of Fit Indices in the model are at acceptable values. According to Table 5 and Figure 2, DL was determined by QMS [$\beta$ = 0.22; S.E: 0.04; t: 5.14; $p < 0.001$; H3a], DL of CM [$\beta$ = 0.31; S.E: 0.05; t: 6.74; $p < 0.001$; H3b] and CM, QMS [$\beta$ = 0.50; S.E: 0.04; t: 13.73; $p < 0.001$; positive direct effects on H3c] were shown. In this respect, a one-unit increase in DL has a 0.22-unit increase on QMS; a one-unit increase in CM indicates an increase of 0.31 in DL and 0.50 in QMS. Similarly, OI of QMS [$\beta$ = 0.35; S.E: 0.04; t: 8.98; $p < 0.001$; H3d] and CM also had OI [$\beta$ = 0.53; S.E: 0.04; t: 14.43; $p < 0.001$; H3e] as positive and moderate direct effects In other words, a one-unit increase in QMS indicates a 0.35 increase in OI, and a one-unit increase in CM indicates a 0.53 increase in OI. The model indicates an indirect effect of CM on QMS via DL [$\beta$ = 0.07; S.E: 0.02; t: 4.08; $p < 0.001$; H3f].

An indirect effect of CM on OI via QMS was also detected in this model [$\beta$ = 0.18; S.E: 0.02; t: 7.55; $p < 0.001$; H3g]. Another indirect specific effect was detected on the OI of CM via the QMS and DL pathway [$\beta$ = 0.02; S.E: 0.01; t: 3.67; $p < 0.001$; H3h].

*4.5. Model 4 Created for Crisis Management, Distance Education, Educational Process and Application Satisfaction, and Organizational Image*

Model 4, which was created for the direct and indirect effects of universities' crisis management skills, distance education and educational process and application satisfaction, and organizational image perceptions according to student perceptions, is given in Table 6.

**Table 6.** Standardized Direct, Indirect and Total Effects on Model 4.

| Hypotheses | Direct Effects | | | Indirect Effect 1 | | | Indirect Effect 2 | | | TE |
|---|---|---|---|---|---|---|---|---|---|---|
| INS→H4a/b/c/d/e/f/g/h | $\beta$ | SE | $p$ * | $\beta$ | SE | $p$ * | $\beta$ | SE | $p$ * | $\beta$ |
| H4a:[DL→INS] | 0.224 | 0.042 | 0.001 * | | | | | | | |
| H4b:[CM→ DL] | 0.308 | 0.046 | 0.001 * | | | | | | | |
| H4c:[CM→INS] | 0.526 | 0.035 | 0.001 * | | | | | | | |
| H4d:[INS→OI] | 0.391 | 0.039 | 0.001 * | | | | | | | |
| H4e:[CM→OI] | 0.495 | 0.037 | 0.001 * | | | | | | | |
| H4f:[CM→DL→INS] | | | | 0.069 | 0.016 | 0.001 * | | | | 595 |
| H4g:[CM→INS→OI] | | | | 0.205 | 0.024 | 0.001 * | 0.027 | 0.007 | 0.001 * | 0.728 |
| H4h:[CM→DL→OI→INS] | | | | | | | | | | |
| MFI | CFI | TLI | RMSE | SRMR | | | $\chi^2$/Sd | | | |
| MFI Values | 0.962 | 0.959 | 0.047 | 0.045 | | | 1390.895/692 = 2.00 | | | |

* $p < 0.001$; MFI: Modification Indices; TE: Total Effect.

It was detected as can be seen in Table 6 that the MFI values of the model showed a Goodness Of Fit Index. According to Table 6 and Figure 2, DL was determined by INS [$\beta$ = 0.22; S.E: 0.04; t: 5.37; $p < 0.001$; H4a], DL of CM [$\beta$ = 0.31; S.E: 0.05; t: 6.74; $p < 0.001$; H4b] and CM, INS [$\beta$ = 0.53; S.E: 0.04; t: 14.99; $p < 0.001$. It appears to have a moderately positive direct effect on H4c]. In this respect, a one-unit increase in DL has a 0.22-unit increase on INS; a one-unit increase in CM indicates an increase of 0.31 in DL and 0.53 in INS. Similarly, OI [$\beta$ = 0.39; S.E: 0.04; t: 10.12; $p < 0.001$; H4d] and CM also have OI [$\beta$ = 0.49; S.E: 0.04; t: 13.37; $p < 0.001$;H4e] as moderate and direct effects in the positive direction. In other words, a one-unit increase in INS indicates a 0.39 increase in OI, and a one-unit increase in CM indicates a 0.49 increase in OI.

In the established model, an indirect effect of CM on INS via DL was detected [$\beta$ = 0.07; S.E: 0.02; t: 4.2; $p < 0.001$; H4f]. Similarly, an indirect effect of CM on OI via INS was achieved [$\beta$ = 0.21; S.E: 0.02; t: 8.46; $p < 0.001$; H4g]. Another indirect specific effect was detected on the OI of CM via the INS and DL pathway [$\beta$ = 0.03; S.E: 0.01; t: 3.83; $p < 0.001$; H4h].

### 4.6. Model 5 Created for Crisis Management, Distance Education, Educational Design Satisfaction, and Organizational Image

Model 5 created for the direct and indirect effects of universities' crisis management skills, distance education, training design satisfaction, and organizational image perceptions according to student opinions is given in Table 7.

**Table 7.** Standardized Direct, Indirect and Total Effects for Model 5.

| Hypotheses | Direct Effects | | | Indirect Effect 1 | | | Indirect Effect 2 | | | TE |
|---|---|---|---|---|---|---|---|---|---|---|
| IDS→H5a/b/c/d/e/f/g/h | β | SE | *p* * | B | SE | *p* * | β | SE | *p* * | β |
| H5a:[DL→IDS] | 0.184 | 0.040 | 0.001 * | | | | | | | |
| H5b:[CM→DL] | 0.307 | 0.046 | 0.001 * | | | | | | | |
| H5c:[CM→IDS] | 0.606 | 0.032 | 0.001 * | | | | | | | |
| H5d:[IDS→OI] | 0.454 | 0.042 | 0.001 * | | | | | | | |
| H5e:[CM→OI] | 0.427 | 0.041 | 0.001 * | | | | | | | |
| H5f:[CM→DL→IDS] | | | | 0.057 | 0.015 | 0.001 * | | | | 0.662 |
| H5g:[CM→IDS→OI] | | | | 0.275 | 0.030 | 0.001 * | | | | 0.728 |
| H5h:[CM→DL→IDS→OI] | | | | | | | 0.026 | 0.007 | 0.001 * | |
| MFI | CFI | TLI | RMSE | SRMR | | | $\chi^2$/Sd | | | |
| MFI Values | 0.941 | 0.937 | 0.058 | 0.050 | | | 1771.95/692 = 2.56 | | | |

\* *p* < 0.001; MFI: Modification Indices; TE: Total Effect.

Table 7 shows that all Goodness Of Fit Indices of the model established were high and appropriate for analysis. When Table 7 and Figure 2 are examined, it is seen that DL has IDS [β = 0.18; S.E: 0.04; t: 4.58; *p* < 0.001; H5a], DL of CM [β = 0.31; S.E: 0.05; t: 6.73; *p* < 0.001; H5b] and CM, IDS [β = 0.61; S.E: 0.03; t: 18.78; *p* < 0.001. It appears to have a moderately positive direct effect on H5c]. In this respect, a one-unit increase in DL has a 0.18-unit increase on IDS; a one-unit increase in CM indicates an increase of 0.31 in DL and 0.61 in IDS. Similarly, OI of IDS [β = 0.45; S.E: 0.04; t: 10.91; *p* < 0.001; H5d] and CM also have OI [β = 0.43; S.E: 0.04; t: 10.34; *p* < 0.001; H5e] as positive and moderate direct effects. In other words, a one-unit increase in IDS indicates a 0.45 increase in OI and a one-unit increase in CM indicates a 0.43 increase in OI.

An indirect effect of CM on IDS over DL was detected in the established model [β = 0.06; S.E: 0.02; t: 3.79; *p* < 0.001; H5f]. Similarly, an indirect effect of CM on OI via IDS was detected [β = 0.28; S.E: 0.03; t: 9.29; *p* < 0.001; H5g]. Another indirect specific effect was detected on the OI of CM via the IDS and DL pathway [β = 0.03; S.E: 0.01; t: 3.55; *p* < 0.001; H5h].

### 4.7. Model 6 Created for Crisis Management, Distance Education, Education, Environment and Resource Satisfaction, and Organizational Image

Model 6, which was created for the direct and indirect effects of universities' crisis management skills, distance education and training design satisfaction, and organizational image perceptions according to student opinions, is given in Table 8.

It is seen in Table 8 that all Goodness Of Fit Indices of the model are high and have appropriate values for analysis. When Table 8 and Figure 2 are examined, it is seen that DL has IES [β = 0.10; S.E: 0.04; t: 2.92; *p* < 0.001; H6a], DL of CM [β = 0.31; S.E: 0.05; t: 6.73; *p* < 0.001; H6b] and CM, IES [β = 0.75; S.E: 0.03; t: 30.58; *p* < 0.001. It appears to have a positive direct effect on H6c]. In this respect, a one-unit increase in DL equals a 0.10-unit increase in IES; a one-unit increase in CM indicates a 0.31 increase in DL and a 0.75 increase in IES. Similarly, OI of IES [β = 0.54; S.E: 0.05; t: 10.11; *p* < 0.001; H6d] and CM also have OI [β = 0.31; S.E: 0.05; t: 5.67; *p* < 0.001; H6e] as positive and moderate direct effects. In other words, a one-unit increase in IES indicates an increase in OI of 0.54, and a one-unit increase in CM indicates a 0.31 increase in OI.

An indirect effect of CM on IES via DL was detected in the established model [β = 0.03; S.E: 0.01; t: 2.7; *p* < 0.001; H6f]. Similarly, an indirect effect of CM on OI over IES was

achieved [β = 0.40; S.E: 0.04; t: 9.32; *p* < 0.001; H6g]. Another indirect specific effect was detected on the OI of CM via the IES and DL pathways [β = 0.02; S.E: 0.01; t: 2.6; *p* < 0.01; H6h].

**Table 8.** Standardized Direct, Indirect and Total Effects for Model 6.

| Hypotheses | Direct Effects | | | Indirect Effect 1 | | | Indirect Effect 2 | | | TE |
|---|---|---|---|---|---|---|---|---|---|---|
| IES→H6a/b/c/d/e/f/g/h | β | SE | *p* * | B | SE | *p* * | β | SE | *p* * | β |
| H6a:[DL→IES] | 0.103 | 0.035 | 0.001 * | | | | | | | |
| H6b:[CM→DL] | 0.307 | 0.046 | 0.001 * | | | | | | | |
| H6c:[CM→IES] | 0.753 | 0.025 | 0.001 * | | | | | | | |
| H6d:[IES→OI] | 0.537 | 0.053 | 0.001 * | | | | | | | |
| H6e:[CM→OI] | 0.307 | 0.054 | 0.001 * | | | | | | | |
| H6f:[CM→DL→IES] | | | | 0.032 | 0.012 | 0.001 * | | | | 0.785 |
| H6g:[CM→IES→OI] | | | | 0.404 | 0.043 | 0.001 * | 0.017 | 0.007 | 0.01 * | 0.728 |
| H6h:[CM→DL→IES→OI] | | | | | | | | | | |
| MFI | CFI | TLI | RMSE | SRMR | | | $\chi^2$/Sd | | | |
| MFI Values | 0.931 | 0.926 | 0.065 | 0.051 | | | 1845.78/619 = 2.98 | | | |

* *p* < 0.001; MFI: Modification Indices; TE: Total Effect.

## 5. Discussion and Conclusions

The present study explained the effects of the crisis management skills of universities and the distance education attitudes of students on their organizational image perception and satisfaction with the structural equation model. When the Goodness of Fit Indices of the structural equation model, which was established [89], which contributes to confirming hypotheses in many fields, e.g., education and analyzing the relations and causality between variables, were examined, it was concluded that it can be used to explain the causal relationship between the variables. The model can explain student satisfaction with sub-dimensions of social and cultural, R&D activity management, monitoring, evaluation and quality management, education and training processes and practices, educational design, educational environment, and resources, and satisfaction with organizational image perception directly or indirectly.

According to the evidence obtained from 467 students who were studying at TRNC state and private universities, all the hypotheses tested by the structural equation model design were supported. The study findings contribute to the reliability of studies reporting similar relations in the literature. Additionally, the emergence of new findings on organizational image and satisfaction in the literature shows the need for a new study to support and generalize the findings obtained. For this reason, related studies were examined from a wider perspective in terms of destination, study group, and techniques used. The findings of the present study were compared and discussed by examining the study results referred to in the relevant literature and the introduction of the study.

It was concluded as a result of the correlation analysis made in the study that there were positive and high-level relations between all variables. This result means that as the crisis management of the university administrations in the COVID-19 pandemic process is perceived positively by the students, the organizational image and satisfaction levels increase. However, it also shows that there is a lower level of relation between attitudes towards distance learning and crisis management, and this has a limited effect on satisfaction with distance education. When the relation between the variables in the relevant literature was reviewed, it was found that when the positive relationship between crisis management and the organizational image was examined theoretically [90,91], it was found that there was no study showing the relations with statistical methods. When the crisis management of universities was evaluated from the perspective of service quality perception, the significant relation between crisis management and student satisfaction can be detected in the literature [18,25,26,28,92] supported by significant relations between distance education

and student satisfaction [37,59,93,94] and the significant relation between organizational image and satisfaction [18,21,95]. The results are compatible with studies in the literature.

In the scope of the first group hypotheses of the study [H1, 2, 3, 4, 5, 6 a, b, c, f], the direct and indirect effects of crisis management on the six dimensions of distance education and student satisfaction were examined. The Structural Equation Model (SEM) analysis showed that distance education has significant direct effects on all six dimensions of student satisfaction, crisis management has significant direct effects on the perception of distance education, and crisis management has significant direct effects on six dimensions of student satisfaction. It was also determined that crisis management has indirect effects on student satisfaction sub-dimensions. The direct effect of distance education on student satisfaction is compatible with the literature data [37,68,93,94,96–101]. Although Buluk and Equalti [96] reported that support services, learning conditions, evaluation systems in distance education, program effectiveness, and students' personal suitability were important determinants of distance education course satisfaction, Atasoy et al. [93] concluded that the e-course implementation processes must be properly planned to increase student satisfaction in the TRNC. Uluskan et al. [101] used the structural equation model and concluded that the design and technical competence of the ESUZEM system is the most important factor affecting student satisfaction. Eygü and Karaman [94] reported that students' course satisfaction was affected by the dimensions of personal suitability, effectiveness, learning, program evaluation, technology, material, evaluation, and support services of distance education and that the highest relation was detected with learning and the lowest relation with technology satisfaction.

In the scope of the second group hypotheses of the study [H1, 2, 3, 4, 5, 6 d, e, g], the direct and indirect effects of crisis management on six dimensions of student satisfaction and organizational image were examined. SEM analysis identified significant direct effects of six dimensions of student satisfaction and crisis management on the organizational image. In addition, crisis management had indirect effects on the organizational image. As a result of the changing perceptions of students about their schools, the messages that universities will send to their students and other stakeholders also changed in a crisis. Right at this point, meeting the demands, needs, and satisfaction of the students by the institution management [102] improves the positive image and increases the resilience against crises [91]. Expressing crisis management as all the planned and conscious activities of an organization not to damage its image and interaction with its stakeholders, Paksoy [103] emphasized the effect of crisis management on the image in his definition. However, expressing that institutions can minimize the damage to their corporate image via public relations activities, Akdağ [90] drew attention to the fact that institutions can make positive contributions to the image of the organization with the success of the managed crisis. Traverso, Román, and González [104] described a university's image according to students; quality of faculty, facilities, physical resources, geographic location, reputation, job placement, market orientation, management, curriculum, number of students in grades, academic plans, student-faculty, relations between students, social services of the university, academic program, evaluation system, and information system for students. Positive image perception also positively affects students' satisfaction levels [21,58,105]. Polat [106], who emphasized product and service quality and responsibilities as the precursor of the organizational image in universities, considered satisfaction as an image output. The innovation competence of educational institution managers emerged as a parameter positively affecting the perception of organizational image [107]. Institutions that can respond quickly to emerging crises, e.g., pandemics with their innovation capabilities will attract the attention of their followers and their care in distance education activities will be welcomed.

The mediated effects between crisis management and organizational image relation were examined in the scope of the final hypothesis group [H1h–H6h] of the study. Although crisis management affects satisfaction via distance education, it also positively affects organizational image via satisfaction. Namely, distance education and satisfaction mediate the way the indirect effect of crisis management affects satisfaction and organi-

zational image. It was found that the study findings were compatible with the literature. Ali et al. [18] examined the effects of service quality of Malaysian public universities on international student satisfaction and corporate image with the Structural Equation Model and reported that all five dimensions of higher education service quality affect student satisfaction and satisfaction also affects the corporate image. Distance education is an important phenomenon for sustainable education both in times of crisis and other times of risk [2]. Chaudhary and Dey [26], who reported that sustainable practices predict student satisfaction, argued that the service quality perceived by students in education had significant effects on their perceptions of the university's sustainable practices and student satisfaction with the structural equation model. As a matter of fact, in his study in which he explained the critical success factors for international education marketing, Mazzarol [108] emphasized that universities' introduction of new teaching techniques via distance education creates an important competitive advantage and image for universities. Universities must consider the distance education practices, which are seen as the understanding of education in crisis periods, as an opportunity for a positive image perception, and must consider the prediction that "even though the pandemic passes, nothing will be the same as before" as a signal of new crises [40].

The results confirm that crisis management, distance education practices, and student satisfaction in the higher education sector are the main motives behind the organizational image. The study emphasizes the role of distance education and student satisfaction as the mediating variables between crisis management and organizational image. This study will also assist university administrations in developing and implementing a market-oriented crisis management strategy to increase student satisfaction, build corporate image, and create a high-quality service in distance education. The study also indicated that the determinants of organizational image and student satisfaction level in education must be better understood, and universities must develop new service plans by reviewing their crisis management skills and distance education practices. Since crisis management and satisfaction have effects on the image, it can be recommended for universities to increase crisis management skills, (i.e., quality of distance learning) and their work to increase satisfaction. We recommend that more crisis management, satisfaction, and image studies must be conducted on service-oriented institutions, e.g., universities.

## 6. Limitations

Despite shedding some light on understanding the direct and indirect relationships among crisis management, student satisfaction, organizational image, and distance learning variables, it is vital to elucidate the limitations of this study to help guide future research. The primary limitation of the present study was that these cross-sectional design results showing the COVID-19 pandemic process have the potential to include change over time. The location of the study (TRNC) and the fact that the participants were mostly students from Turkey and TRNC are thought to have the potential to limit its generalizability to other countries and cultures. Right at this point, it is necessary to be supported by studies in different countries and cultures. Variables of satisfaction and attitude must be taken into account in limitations because they are related to many respects such as the socioeconomic structures of the individual and countries. This study was only conducted at ten public and private universities in the TRNC. Thus, as the sample size was not large, the results from this study cannot be generalized to the wider population of higher education students in all other countries. We suggest that similar studies in other public and private universities in other countries and/or in other cultures can be conducted to provide more fruitful insights and extend the generalizability of the findings to annihilate or at least decrease the endogenous selection bias. Moreover, this study only adapted related research variable dimensions to assess the potential effects of crisis management skills of TRNC universities on student satisfaction and organizational image via distance learning during the COVID-19 pandemic; future research might consider comparative and longitudinal studies between face-to-face and distance learning education that may affect student

satisfaction and organizational image. Finally, as pointed out by a number of scholars, this study is limited to relying on observational data sets and regression models where variables cannot be exogenously manipulated [109]. The current research is also limited to the assumption that the variables meet the three sources of endogeneity bias caused by possible measurement error, simultaneity, and omitted variables [110,111] which could render the coefficient estimates from standard regressions to be causally uninterpretable.

**Author Contributions:** Conceptualization, S.T. and F.S.; methodology, Ü.K., E.T. and S.T.; software, E.T. and Ü.K.; validation, R.A., S.T. and F.S.; formal analysis, Ü.K. and R.A.; investigation, S.T., F.S. and R.A.; resources, Ü.K. and E.T.; data curation, Ü.K. and R.A.; writing—original draft preparation, E.T. and Ü.K.; writing—review and editing, S.T., F.S. and E.T.; visualization, E.T.; supervision, S.T. and F.S.; project administration, F.S.; funding acquisition, E.T. All authors have read and agreed to the published version of the manuscript.

**Funding:** This research received no external funding.

**Institutional Review Board Statement:** Approval was obtained from the Cyprus International University Ethics Committee for this research: 100-3364-19.04.2021.

**Informed Consent Statement:** Informed consent was obtained from all subjects involved in the study.

**Data Availability Statement:** Data available upon request.

**Conflicts of Interest:** The authors declare no conflict of interest.

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
