# Peer review of "The Effects of the Crisis Management Skills and Distance Education Practices of Universities on Student Satisfaction and Organizational Image"

_sustainability, doi:10.3390/su14105813_

Round 1

Reviewer 1 Report

Very actual topic, middle-size sample, good quality of text, excelent literature review, excelent discussion of results

Minor comments:

page 3, lines 109 - 110. You should provide more details about works [13] - [19]
page 5, lines 248 - 252.  You should provide more details about works [48] - [60]
page 7, lines 304, 306 and 308. It would be better H1, H2, H3, ...

Author Response

Response to Reviewer 1 Comments

First of all, thank you for reviewing my research article and for your comments. I share my answers and arrangements for your comments.

Point 1: “Page 3, lines 109 - 110. You should provide more details about works [13] - [19]”

Response 1: In line with the referee's comments, more details are given about the related works. [in line: 104-127]

Point 2: “Page 5, lines 248 - 252.  You should provide more details about works [48] - [60]”

Response 2:  In line with the referee's comments, more details are given about the related works. [in line: 262-277]

Point 3: “Page 7, lines 304, 306 and 308. It would be better H1, H2, H3, ...”

Response 3: Changes were made in line with the referee's comments. [in line: 331-333-335]

If you wish, I would be happy to answer your questions about the answers I gave to your comments. Thanks

Reviewer 2 Report

The article addresses a topic of great interest to the magazine's readers. It presents a good review of the literature and a research approach in accordance with the previously established objectives.

Author Response

Response to Reviewer 2 Comments

First of all, thank you for reviewing my research article and for your comments. I share my answers and arrangements for your comments.

 Response: In the context of contributing to the content and theoretical infrastructure of the research, the literature has been re-examined, and research findings that will contribute to the clear presentation of the results for empirical research and support the arguments in the findings are included.

If you wish, I would be happy to answer your questions about the answers I gave to your comments. Thanks

Reviewer 3 Report

  1. This article fails to identify a clear research gap.
  2. It also fails to contextualize this study in international context. Thus, it is not clear whether the results are critical challenges for the certain country or other countries.
  3. The method used in this study fails to account for multiple endogenous self-selection issues, which could lead to biased estimates.
  4. The research design is not adequate for inferring whether there is a causal relationship between Student Satisfaction via Crisis Management Skills.

Author Response

Response to Reviewer 3 Comments

First of all, thank you for reviewing my research article and for your comments. I share my answers and arrangements for your comments.

Point 1: “This article fails to identify a clear research gap.”, “ It also fails to contextualize this study in international context. Thus, it is not clear whether the results are critical challenges for the certain country or other countries.”, “The method used in this study fails to account for multiple endogenous self-selection issues, which could lead to biased estimates.”, “The research design is not adequate for inferring whether there is a causal relationship between Student Satisfaction via Crisis Management Skills.”

Response 1: In line with the comments of the referee, the limitations section of the article has been detailed. [in line: 296-298, 691-718]

If you wish, I would be happy to answer your questions about the answers I gave to your comments. Thanks

Reviewer 4 Report

The study investigates the phenomenon of whether the quality of (pandemic) crisis management of a higher education institution perceived by students is connected to student satisfaction and the image of the organization. This question is of little scientific interest, as it is difficult to assume that this would not be the case. Another serious limitation against generalizability is that the empirics are based on a non-representative sample from Northern Cyprus only. On the other hand, empirical testing is still important.
The SEM analysis is correct, it is the strength of the manuscript. However, there are several weaknesses as well (listed below).
The authors mention that they search the literature but it should be reported what literature sources were scanned and how (lines 266-267).
The literature review is far too long considering that it contains a lot of information loosely connected to the topic. It should be shorter and more focused on the research questions.
The definition of variables and dimensions should be given before they are placed in Figure 1 and mentioned in the Study questions. Without it, the reader has no chance to understand the information in the figure and what sub-dimensions the questions are addressing. 
The study questions should be more explicitly connected to the literature review.
The description of the sampling is missing (how were the ten universities selected, how were the students selected), as well as a detailed description of the data collection process. The authors do not inform us appropriately on how did they ensure voluntariness (line 332).  
The English of the study is very poor, it even hinders the understanding several times. 

Minor comments:
In line 40, the education sector is described as particularly affected. It would be worth mentioning other strongly affected sectors (like hospitality and tourism).
Short term goals are affected as well (line 112).
The sources of data between lines 79-85 are missing.
Several sentences are not understandable (like in lines 26, 98-99, 108-110, 124-126…).
Unreasonable duplication of words: lines 148-149, 609-610.
Table 2 would be clearer if the variables were identified by their names on the horizontal axis, too. The size of it could be decreased by omitting the column CM (or 1) because it is not containing any information.
The population is surely higher than the number of HEIs (line 323). It could be a mistake in the sentence.

Author Response

Response to Reviewer 4 Comments

First of all, thank you for reviewing my research article and for your comments. I share my answers and arrangements for your comments.

Point 1: “The authors mention that they search the literature but it should be reported what literature sources were scanned and how (lines 266-267).”

Response 1: The relevant section has been rearranged in line with the referee's comments. [in line: 291-294]

Point 2: “The definition of variables and dimensions should be given before they are placed in Figure 1 and mentioned in the Study questions. Without it, the reader has no chance to understand the information in the figure and what sub-dimensions the questions are addressing.”

Response 2:  Relevant arrangements have been made so that the reader can better understand the information and sub-dimensions in the figure 1. [in line: 302-309]

Point 3: “The description of the sampling is missing (how were the ten universities selected, how were the students selected), as well as a detailed description of the data collection process. The authors do not inform us appropriately on how did they ensure voluntariness (line 332).  ”

Response 3: Information about the universities, students and data collection process in the research sample was added by rearranging. [in line: 359-367]

Point 4: “In line 40, the education sector is described as particularly affected. It would be worth mentioning other strongly affected sectors (like hospitality and tourism).”

Response 4: The sentence was rearranged in line with the referee's comment. [in line: 36-40]

Point 5: “Short term goals are affected as well (line 112).”

Response 5: In line with the referee's comments, the word has been added. [in line: 129]

Point 6: The sources of data between lines 79-85 are missing.

Response 6: The source has been added in line with the referee's comments. [in line: 86]

Point 7: “Several sentences are not understandable (like in lines 26, 98-99, 108-110, 124-126…).

Response 7: The sentences were revised in line with the referee's comments. [in line: 26-27, 99-100,124-127, 140-143]

Point 8: Unreasonable duplication of words: lines 148-149, 609-610.

Response 8: The sentences were rearranged in line with the referee's comments. [in line: 163-166, 646-651]

Point 9: Table 2 would be clearer if the variables were identified by their names on the horizontal axis, too. The size of it could be decreased by omitting the column CM (or 1) because it is not containing any information.

Response 9: Table 2 has been rearranged in line with the referee's comments. [in line: 435]

Point 10: The population is surely higher than the number of HEIs (line 323). It could be a mistake in the sentence.

Response 10: The sentence was rearranged in line with the referee's comments. [in line: 349-352]

If you wish, I would be happy to answer your questions about the answers I gave to your comments. Thanks
